



# UBER v1.0: A universal kinetic equation solver for radiation belts

Liheng Zheng[1], Lunjin Chen[1], Anthony A. Chan[2], Peng Wang[3], Zhiyang Xia[1], and Xu Liu[1]

[1]William B. Hanson Center for Space Sciences, Department of Physics, University of Texas at Dallas, Richardson, Texas, USA.
[2]Department of Physics and Astronomy, Rice University, Houston, Texas, USA.
[3]Department of Earth, Planetary and Space Sciences, University of California at Los Angeles, Los Angeles, California, USA.

**Correspondence:** Liheng Zheng (zhengliheng@gmail.com)

**Abstract.** Recent proceedings in the radiation belt studies have proposed new requirements for numerical methods to solve the kinetic equations involved. In this article, we present a numerical solver that can solve the general form of radiation belt Fokker-Planck equation and Boltzmann equation in arbitrarily provided coordinate systems, and with user-specified boundary geometry, boundary conditions, and equation terms. The solver is based upon the mathematical theory of stochastic differential equations, whose computational accuracy and efficiency are greatly enhanced by specially designed adaptive algorithms and variance reduction technique. The versatility and robustness of the solver is exhibited in three example problems. The solver applies to a wide spectrum of radiation belt modeling problems, including the ones featuring non-diffusive particle transport such as that arises from nonlinear wave-particle interactions.

## 1 Introduction

In space plasma environment, the radiation belts refer to torus-shaped regions surrounding Earth and other magnetized planets that are filled with highly energetic charged particles trapped in the planetary magnetic field. Since their discovery (Van Allen and Frank, 1959; Vernov et al., 1959), the radiation belts have been the focus of intense research due to the innumerable unknowns concerning their extremely dynamic behavior and their damaging effects on spacecraft (e.g., Baker, 2000; Welling, 2010). During slowly changing conditions, radiation belt particles undergo three types of periodic motion: gyration about field lines, bounce along field lines, and drift about the planet. With each periodic motion there is a corresponding adiabatic invariant, defined through Hamiltonian action integral, that is only violated when the conditions are changing on time scales shorter than the period. A widely adopted method to study the dynamics of radiation belts is to solve a kinetic equation describing the evolution of particle phase space density. In quasi-linear theory, this kinetic equation is usually a Fokker-Planck equation that takes the general covariant form (Schulz, 1991)

$$\frac{\partial \bar{f}}{\partial t} = \frac{1}{G} \frac{\partial}{\partial Q^\alpha} \left( G D^{\alpha\beta} \frac{\partial \bar{f}}{\partial Q^\beta} \right) - \frac{1}{G} \frac{\partial}{\partial Q^\alpha} \left( G h^\alpha \bar{f} \right) + S \bar{f} + v, \tag{1}$$

where $\bar{f}$ is the phase-averaged phase space density, $G = \det(\frac{\partial J^\alpha}{\partial Q^\beta})$ is the Jacobian determinant for the transformation from canonical action-integral variables $J^\alpha$ ($\alpha = 1, 2, 3$) to the generalized coordinates $Q^\alpha$, and $D^{\alpha\beta}$, $h^\alpha$, $S$ and $v$ are coefficients of the equation. Summation on repeated Greek indices is implied throughout this paper. In different radiation belts, the number





of terms emerging on the right-hand side of Eq. (1) and their respective physical backgrounds may be different. For the Earth's outer radiation belt, the second and the fourth terms are usually missing; the first term represents diffusion caused by wave-particle interactions, and the third term is often a loss characterized by the particle lifetime (e.g., Li and Hudson, 2019, and the reference therein). In the low-altitude inner radiation belt where wave-particle interactions are not as significant, the first and the second terms are often provided by the diffusion and dynamic friction caused by inter-particle Coulomb collisions (e.g., Selesnick, 2012), and the fourth term may be a source from cosmic ray albedo neutron decay (CRAND) (e.g., Selesnick, 2015; Li et al., 2017). For radiation belts of the gas giants, all terms could be present (e.g., Horne et al., 2008; Lorenzato et al., 2012). The first two terms may be attributable to both wave-particle interactions and inter-particle collisions, and in addition, synchrotron radiation, which is negligible in Earth's radiation belts, bleeds energy for the ultra-relativistic electrons and thus also contributes to the second term (e.g., Bolton et al., 2002, 2004). The third term could represent the moon-sweeping loss, and the fourth term may come from moon volcanic activities as a plasma source (e.g., Nénon et al., 2017).

In some circumstances, the dependence of phase space density on certain phases $\varphi^\iota$ could be discerned, and the radiation belt kinetic equation takes the form

$$\frac{\partial \bar{f}}{\partial t} + \dot{\varphi}^\iota \frac{\partial \bar{f}}{\partial \varphi^\iota} = \frac{1}{G} \frac{\partial}{\partial Q^\mu} \left( G D^{\mu\nu} \frac{\partial \bar{f}}{\partial Q^\nu} \right) - \frac{1}{G} \frac{\partial}{\partial Q^\mu} \left( G h^\mu \bar{f} \right) + S\bar{f} + v, \tag{2}$$

where a dot over $\varphi^\iota$ indicates its time derivative. The phase space density $\bar{f}$ here is only averaged over the phases varying faster than $\varphi^\iota$, and the mutually exclusive indices $\iota$ and $\mu$ together form the complete set of $\alpha$. The most common situation is perhaps the dependence of $\bar{f}$ on the drift phase $\varphi^3$, which in the Earth's outer radiation belt may be caused by the wave activity dependence on magnetic local time (e.g., Shprits et al., 2009), and in the inner belt by the longitudinal variation of drift shell altitude (e.g., Tu et al., 2010; Xiang et al., 2019). With the spatial derivative term on the left-hand side, Eq. (2) appears as a Boltzmann equation for $\bar{f}$; but by Hamiltonian mechanics, the conjugating term $\dot{Q}^\iota (\partial \bar{f}/\partial Q^\iota)$ should have also appeared on the left-hand side. Its absence is due to the fact that, for particles in the radiation belt energy range, the drift-phase-dependent electric potential energy is usually negligible in the unperturbed particle Hamiltonian, so that $\varphi^3$ becomes a cyclic variable. Appendix A provides a more comprehensive explanation on this equation; and we will return to the general case where $\varphi^3$ is not cyclic in the discussion section.

Various numerical models have been built to solve a specific form of either Eq. (1) or Eq. (2) (e.g., Beutier et al., 1995; Selesnick et al., 2003; Tao et al., 2008; Albert et al., 2009; Subbotin et al., 2010; Tu et al., 2013; Wang et al., 2017; Xiang et al., 2020, to name a few), and each of them is built with a hard-coded choice of coordinates, relatively simple boundary geometry, and roughly fixed number of equation terms; therefore, each model is only applicable to a specific set of problems. This situation could become quite inconvenient when adiabatic invariants of particle motion are used as coordinates of phase space to model radiation belt dynamics, as promoted by Schulz (1996). The reasons are two-fold: first, due to their vast range of magnitude and dramatically varying resolution, adiabatic invariant coordinates often require some kind of rescaling and transformation (e.g., Zheng et al., 2014), specific to the problem, to be computationally efficient; and second, boundary geometry becomes complicated in adiabatic invariant coordinates, which could be challenging for numerical methods and had led Subbotin and Shprits (2012) to seek for new coordinates from combinations of the adiabatic invariants. However, the use





of adiabatic invariant coordinates is crucial for some compelling problems in the radiation belts, for example the mechanisms of storm-time electron loss in which adiabatic modulations due to magnetic field configuration change must be separated from non-adiabatic processes (Kim and Chan, 1997; Turner and Ukhorskiy, 2020), and the relative significance of Earthward diffusion versus CRAND as possible inner belt electron source where drift shell splitting effect contributes (Cunningham et al.,
2018). It is the purpose of this article to present a numerical code, named UBER (for "universal Boltzmann equation solver"), that solves Eq. (1) and Eq. (2) in an arbitrarily user-specified coordinate system up to three dimensions, with great freedom in specifying boundary geometry and boundary conditions, and with various combinations of equation terms. Therefore, it is expected that the solver can be applied to a wide spectrum of radiation belt modeling problems. More importantly, the freedom of specifying equation terms implies that, in an asymptotic manner, UBER can even solve the integro-differential
kinetic equations arising from non-diffusive particle transport, such as that formulated in Artemyev et al. (2018) for nonlinear wave-particle interactions, and thereby provides a viable means to incorporate non-diffusive transport into global radiation belt modeling.

The underlying mathematical theory of the solver is stochastic differential equation (SDE) theory. The SDE method had been utilized by Tao et al. (2008), Selesnick et al. (2013) and Zheng et al. (2014) in their modeling of the radiation belts. The
method is grid-free, and enjoys unparalleled advantages in dealing with cross diffusion components and complicated boundary geometry (e.g., Zheng et al., 2016), but is meanwhile notorious for low efficiency ascribed to its Monte Carlo nature. In this article, we also describe specially designed numerical techniques that have enhanced the computational speed of the SDE method by an order of magnitude, thus making the solver much affordable to large-scale simulations. Three example problems with distinct physical backgrounds are provided in this article to demonstrate the abilities and versatility of the solver.

## 75  2   Mathematical Theory

The kinetic equations (1) and (2) are parabolic partial differential equations (PDEs). Written in the Kolmogorov backward form (see below), a parabolic PDE corresponds to a multi-dimensional SDE that describes the motion of an Itô stochastic process whose certain functional expectation satisfies the PDE; and the PDE can then be solved by calculating path integrals of the corresponding stochastic process (e.g., Freidlin, 1985; Øksendal, 1998).
Let us consider the following partial differential problem composed of a Kolmogorov backward equation and a general set of initial and boundary conditions:

$$\partial_t f = \frac{1}{2} a^{\alpha\beta}(t, \boldsymbol{x}) \partial_\alpha \partial_\beta f + b^\alpha(t, \boldsymbol{x}) \partial_\alpha f + c(t, \boldsymbol{x}) f + u(t, \boldsymbol{x}), \tag{3}$$

$$f(0, \boldsymbol{x}) = g_0(\boldsymbol{x}), \quad \boldsymbol{x} \in \bar{\Omega}, \tag{4}$$

$$f(t, \boldsymbol{x}) = g_1(t, \boldsymbol{x}), \quad \boldsymbol{x} \in \partial_1 \Omega, \tag{5}$$

$$\hat{\boldsymbol{\gamma}}(\boldsymbol{x}) \cdot \nabla f - \lambda(t, \boldsymbol{x}) f = 0, \quad \boldsymbol{x} \in \partial\Omega \setminus \partial_1 \Omega, \tag{6}$$

where $\partial_t$ and $\partial_\alpha$ are shorthands for the partial differentials with respect to $t$ and the $\alpha$-th coordinate, respectively. In Eqs. (4)-(6), $\bar{\Omega}$ denotes the closure of the domain and $\partial\Omega$ its boundary. In particular, $\partial_1\Omega$ are the boundary pieces of the first type





(Dirichlet) boundary condition, and $\partial\Omega \setminus \partial_1\Omega$ indicate the boundary pieces excluding those in $\partial_1\Omega$, which are of the second

(Neumann, $\lambda \equiv 0$) or the third type (Robin, $\lambda \neq 0$) boundary conditions. The unit vector $\hat{\gamma}$ points into $\bar{\Omega}$ and is not tangent to the local boundary.

The mathematical theory of SDEs establishes a relation between Eqs. (3)-(6) and the Itô stochastic process, whose spatial positions are denoted by the random variable $\boldsymbol{X}_s$ in $\Omega$, that obeys the reflected SDE

$$d\boldsymbol{X}_s = \boldsymbol{b}(t-s, \boldsymbol{X}_s)ds + \boldsymbol{\sigma}(t-s, \boldsymbol{X}_s) \cdot d\boldsymbol{W}_s + \hat{\boldsymbol{\gamma}}(\boldsymbol{X}_s)dk_s, \tag{7}$$

where the dot product on the right-hand side is between a rank-2 tensor and a vector, and the parameter $s$ runs from 0 to $t$, so

that the stochastic process retrogrades in time from $t$ to 0. The first term on the right-hand side describes the ballistic part of its motion. The second term describes the stochastic part, with the coefficient tensor $\boldsymbol{\sigma}$ satisfying $\boldsymbol{\sigma} \cdot \boldsymbol{\sigma}^T = \mathbf{a}$ (whose components are $a^{\alpha\beta}$). Note that this condition does not uniquely determine $\boldsymbol{\sigma}$, but all satisfying $\boldsymbol{\sigma}$'s are equivalent (Levi's theorem, Freidlin, 1985; Zheng et al., 2014). $\boldsymbol{W}_s$ is a vector Wiener process of the same dimensions as $\boldsymbol{X}_s$, with each dimension an independent Gaussian stochastic variable that has zero mean and variance $s$. The third term describes reflection of the stochastic process

in the direction given by $\hat{\boldsymbol{\gamma}}$ on the boundary $\partial\Omega \setminus \partial_1\Omega$, and $k_s$ is a monotonic stochastic variable that only increases when the stochastic process is on that boundary to force $\boldsymbol{X}_s$ to stay in $\bar{\Omega}$. $k_s$ can thus be considered as a measure of the time that the stochastic process spent on $\partial\Omega \setminus \partial_1\Omega$, and hence has the name local time. The Itô process stops either in $\bar{\Omega}$ when $s = t$, or on $\partial_1\Omega$ at $s = \tau < t$.

A formal solution of the problem in Eqs. (3)-(6) is given by the Feynman-Kac formula (e.g., Kac, 1949; Øksendal, 1998;

Klebaner, 2005)

$$f(t, \boldsymbol{x}) = \mathbb{E}\big[\mathcal{F}^{t,\boldsymbol{x}}[\boldsymbol{X}_s]\big], \tag{8}$$

in which $\mathbb{E}$ is the expectation operator, and $\mathcal{F}^{t,\boldsymbol{x}}[\boldsymbol{X}_s]$ is a functional of the stochastic path $\boldsymbol{X}_s$ started from $t$ and $\boldsymbol{x}$, and has the expression

$$\begin{aligned}
\mathcal{F}^{t,\boldsymbol{x}}[\boldsymbol{X}_s] = {}& \mathbb{I}_{\tau \geq t}\, g_0(\boldsymbol{X}_t) \exp\left[\int_0^t c(t-s, \boldsymbol{X}_s)ds - \int_0^t \lambda(t-s, \boldsymbol{X}_s)dk_s\right] \\
& + \mathbb{I}_{\tau < t}\, g_1(t-\tau, \boldsymbol{X}_\tau) \exp\left[\int_0^\tau c(t-s, \boldsymbol{X}_s)ds - \int_0^\tau \lambda(t-s, \boldsymbol{X}_s)dk_s\right] \\
& + \int_0^{t \wedge \tau} u(t-s, \boldsymbol{X}_s) \exp\left[\int_0^s c(t-r, \boldsymbol{X}_r)dr - \int_0^s \lambda(t-r, \boldsymbol{X}_r)dk_r\right]ds,
\end{aligned} \tag{9}$$

where the symbol $\mathbb{I}_{\tau \geq t}$ is equal to one when $\tau \geq t$, which means the stochastic process has stopped in $\bar{\Omega}$ before it had a chance to reach $\partial_1\Omega$, and zero otherwise; and $t \wedge \tau$ means the smaller between the two. Physically, the functional $\mathcal{F}^{t,\boldsymbol{x}}[\boldsymbol{X}_s]$ is a

propagator of contribution carried along the stochastic path from either the initial condition or the first type boundary condition to the point of solution, and the exponential functions indicate how this contribution enhances or decays along this path.

To formally solve the Fokker-Planck equation (1) by the Feynman-Kac formula (8), it remains to transform the equation together with its proper initial and boundary conditions into the form of Eqs. (3)-(6). To this end, directly expanding Eq. (1)





and collecting terms with the same differentiation order yields its Kolmogorov backward form

$$\partial_t \bar{f} = D^{\alpha\beta}\partial_\alpha\partial_\beta\bar{f} + \left[(\partial_\beta D^{\alpha\beta} + D^{\alpha\beta}\partial_\beta \ln G) - h^\alpha\right]\partial_\alpha\bar{f} + \left[S - (\partial_\alpha h^\alpha + h^\alpha\partial_\alpha \ln G)\right]\bar{f} + v. \tag{10}$$

Comparing Eq. (10) with Eq. (3) and taking $Q^\alpha$ equivalent to $x^\alpha$, we thus have the correspondences of coefficients that:

$$\begin{cases} \mathbf{a} = 2\mathbf{D}, \\ \boldsymbol{b} = \nabla\cdot\mathbf{D} - \boldsymbol{h}, \\ c = S - \nabla\cdot\boldsymbol{h}, \\ u = v, \end{cases} \tag{11}$$

where in curvilinear coordinates, the divergence operator on a tensor field $\boldsymbol{\Gamma}$ is

$$\nabla\cdot\boldsymbol{\Gamma} = \partial_\alpha\Gamma^{\alpha\cdots} + \Gamma^{\alpha\cdots}\partial_\alpha \ln G, \tag{12}$$

in which the dots stand for all other indices irrelevant to the operation, and the terms $\partial_\alpha \ln G$ come from summation of the Christoffel symbols in a covariant derivative (e.g., Mathews and Walker, 1970, Chap. 15). It is worth remarking that $-\boldsymbol{h}$ appears in the expression for $\boldsymbol{b}$, so that the Itô process travels against the advection velocity. This is indeed the case since it is

time-backwards. Also, from the expression for $c$, divergence of the advection serves as a loss of phase space density.

Initial and boundary conditions to Eq. (1) are transformed as follows. For the initial condition and the first type boundary condition, values of $\bar{f}(t, \boldsymbol{x})$ are specified just as in Eqs. (4) and (5). For a flux boundary condition of the form $\Phi = g_2(t, \boldsymbol{x})\bar{f}$, we note that the outward flux $\Phi$ across a boundary is given by $(\hat{\boldsymbol{n}}\cdot\mathbf{D}\cdot\nabla\bar{f} - \hat{\boldsymbol{n}}\cdot\boldsymbol{h}\bar{f})$, with $\hat{\boldsymbol{n}}$ the unit inward normal vector of $\partial\Omega \setminus \partial_1\Omega$. Therefore, the corresponding boundary condition is

$$\hat{\boldsymbol{n}}\cdot\mathbf{D}\cdot\nabla\bar{f} - (\hat{\boldsymbol{n}}\cdot\boldsymbol{h} + g_2)\bar{f} = 0. \tag{13}$$

Comparing Eq. (13) with Eq. (6), we identify that:

$$\begin{cases} \hat{\boldsymbol{\gamma}} = \dfrac{\hat{\boldsymbol{n}}\cdot\mathbf{D}}{|\hat{\boldsymbol{n}}\cdot\mathbf{D}|}, \\ \lambda = \dfrac{\hat{\boldsymbol{n}}\cdot\boldsymbol{h} + g_2}{|\hat{\boldsymbol{n}}\cdot\mathbf{D}|}. \end{cases} \tag{14}$$

Although the SDE (7) does not prevent $\boldsymbol{\sigma}$, and hence $\mathbf{D}$, from being zero, the expressions in Eqs. (14) do become singular for vanishing $\mathbf{D}$ on $\partial\Omega \setminus \partial_1\Omega$. In the region where $\mathbf{D}$ vanishes, Eq. (3) is no longer parabolic but degenerates to an advection equation (a first order PDE), for which imposing a Neumann or Robin boundary condition is over-determinant. In this case, we invoke on the boundary minimal diffusion in the eigen-direction of $\hat{\boldsymbol{n}}$ so that $\hat{\boldsymbol{\gamma}} = \hat{\boldsymbol{n}}$, and let $g_2 \equiv -\hat{\boldsymbol{n}}\cdot\boldsymbol{h}$ so that $\lambda = 0$,

which means the advective flow is free to cross the boundary. The situation that $\mathbf{D}$ is finite but $|\hat{\boldsymbol{n}}\cdot\mathbf{D}|$ vanishes is considered pathological to our problem.

Up to this point, we have transformed the Fokker-Planck equation (1) and its initial and boundary conditions to the problem in Eqs. (3)-(6), and gathered all expressions in Eqs. (11) and (14) for the constructing components of the SDE (7) as well as the functional (9). In order to solve the Boltzmann equation (2), it suffices for us to just transform the equation into the form





**Table 1.** User input items to the UBER code

| Input items | Comments |
|---|---|
| $\partial_\alpha \ln G$ | Vector field to specify the coordinate system |
| $D^{\alpha\beta}, h^\alpha, S, v$ | Coefficients to define the PDE |
| $g_0(x^\alpha)$ | Function to provide the initial condition |
| $\psi(t, x^\alpha) = 0$ | Equation to define a boundary piece's geometry |
| $g_*(t, x^\alpha)$ | Function to provide the boundary condition, |
| | $* = 1$ or $2$ depending on the type of the boundary |
| $\hat{\boldsymbol{n}}(x^\alpha)$ | Inward unit normal vector only for $\partial\Omega \setminus \partial_1\Omega$ |

A set of the boundary-related items for each piece of boundary.

of Eq. (1). To this end, we expand the phase space by concatenating the coordinates $Q^\mu$ and $\varphi^\iota$, so that $x^\alpha = \{Q^\mu, \varphi^\iota\}$ (recall that $\alpha = \{\mu, \iota\}$), and introduce the new coefficients $\widetilde{D}^{\alpha\beta}$, $\widetilde{h}^\alpha$, $\widetilde{S}$ and $\widetilde{v}$ that satisfy the following conditions:

$$
\begin{cases}
\widetilde{D}^{\mu\nu} = D^{\mu\nu}, \quad \widetilde{D}^{\alpha\iota} = 0, \\
\widetilde{h}^\alpha = \{h^\mu, \dot{\varphi}^\iota\}, \\
\widetilde{S} = S + \partial_\iota \dot{\varphi}^\iota, \\
\widetilde{v} = v.
\end{cases}
\tag{15}
$$

It can be verified that Eq. (1) in the new $x^\alpha$ coordinates with the new coefficients given by (15) transforms into Eq. (2) after replacing $x^\alpha$ by $Q^\mu$ and $\varphi^\iota$. The transformation (15) essentially treats $\varphi^\iota$ as new dimensions of the stochastic motion, except that the stochastic part of the motion in these dimensions is identically zero. A new type of boundary condition might emerge

for problems involving Eq. (2), that is the periodic boundary condition for the phases $\varphi^\iota$. From the viewpoint of stochastic motion, though, such periodicity is not really a boundary but rather a topology of $\Omega$. The treatment of periodic boundary condition will be exemplified in the third problem in Section 4 below.

To summarize this section, the above mathematical theory allows us to fully define a PDE problem involving Eq. (1) or Eq. (2) in an arbitrary coordinate system given the input functions and equations as listed in Table 1, which can be either

analytical or numerical in the UBER code. The equation terms may be freely turned off by setting their corresponding coefficients zero. The number of boundary pieces is totally up to choice, which can even be zero to put the boundary at infinity. The boundary geometry may be time-variable for boundary pieces in $\partial_1\Omega$, but must be fixed for those in $\partial\Omega \setminus \partial_1\Omega$. Solutions of this problem are obtained once we find a way to evaluate the functional in Eq. (9) for a realization of a stochastic path, and to estimate the expectation of the functional. These numerical techniques are the subject of the next section.

## 3 Numerical Techniques

We give an outline of the algorithms used by the UBER code in this section, with emphasis on the techniques that improve both its accuracy and efficiency. Lower-level numerical techniques, such as the generation of pseudo-random variables, linear





algebraic operations, and parallelized computation, are based on the works presented in Zheng (2015). The general idea for numerically implementing the SDE method is as follows: (i) for a given spatiotemporal position $(t, \boldsymbol{x})$ where an equation

solution is wanted, a number of stochastic paths starting from this common position are simulated; (ii) for each stochastic path, its functional value is evaluated by the path integrals as in Eq. (9); and (iii) from these sampled functional values, their expectation is estimated, and this gives the solution at $(t, \boldsymbol{x})$. Therefore, the SDE method is essentially a Monte Carlo method. It does not rely on a computational grid, and is able to solve the problem locally. However, in many occasions it is still worth obtaining global solutions on a grid, so that the solutions at time stamp $T_i$ may be used as the initial condition for the

solutions at $T_{i+1}$, analogous to the idea of the layer methods (e.g., Tao et al., 2009). In this way, the stochastic processes need only to be simulated for a short duration of $t = T_{i+1} - T_i$ to obtain the new solutions, for which the calculation of functional expectation would converge much faster than those simulated for the full length $t = T_{i+1}$. The only operation on this grid would be interpolation and possibly extrapolation, therefore unlike in the layer methods, the grid needs not to be uniform or even regular.

Integration of the SDE (7) employs the Euler-Maruyama scheme that is order 1 for weak convergence problems such as ours, meaning that when only the statistical distribution of stochastic paths matters but not the individual path, the expectation of the schematic error is proportional to the first power of the time stepsize (Kloeden and Platen, 1992). To further reduce the schematic error, an adaptive time stepsize is used in UBER. It can be shown that (e.g., Zheng, 2015) the root-mean-square (RMS) distance an Itô stochastic process travels in infinitesimal time $ds$ is

$$d\bar{X}_s = \sqrt{\mathrm{tr}(\mathbf{a})ds}. \tag{16}$$

Numerically, the first order contribution from $|\boldsymbol{b}\Delta s|$ cannot be neglected due to the finite $\Delta s$. Therefore, we prescribe a desired RMS spatial stepsize $\Delta\bar{X}_s$, which is sufficiently small compared to the size of $\Omega$ and any scale length of the equation coefficients, and then choose the smaller $\Delta s$ inferred from either Eq. (16) or $\Delta\bar{X}_s = |\boldsymbol{b}\Delta s|$ at every step of integration as the adaptive stepsize. This scheme evidently reduces to a simple adaptive Euler scheme for integrating ordinary differential equations when $\mathbf{a}$ approaches zero.

Oblique reflection of the stochastic process on $\partial\Omega \setminus \partial_1\Omega$ and the calculation of $dk_s$ follow the projected-half-space algorithm presented in Gobet (2001), which is also order 1 in the weak convergence sense. The idea is that, for an exact half-space boundary, $k_s$ can be proven to share the same probabilistic distribution with a composite stochastic variable involving $\boldsymbol{W}_s$, coefficients of the SDE, the normal vector $\hat{\boldsymbol{n}}$, and an independent exponential random variable with parameter $(2s)^{-1}$ (Lépingle, 1995), and therefore $k_s$ can be explicitly calculated by these known quantities. For general smooth boundary geometry, an

additional contribution to $dk_s$ may also come from possible projection along the $\hat{\boldsymbol{\gamma}}$ direction needed to keep the stochastic process within domain. With $dk_s$ obtained and the SDE (7) integrated, the functional (9) can be readily evaluated by an ordinary numerical integration technique implemented along the realized stochastic path.

    Expectation of the functionals can be estimated, in principle, from an arithmetic mean of a number $N$ of sampled stochastic path integrals. The error of this estimation, $\epsilon = |\mathbb{E}[\mathcal{F}^{t,\boldsymbol{x}}] - \langle\widetilde{\mathcal{F}}^{t,\boldsymbol{x}}\rangle|$ where a tilde is used to indicate a numerical realization in

this section and $\langle\cdots\rangle$ indicates averaging over samples, can be estimated by dividing the simulation of stochastic processes





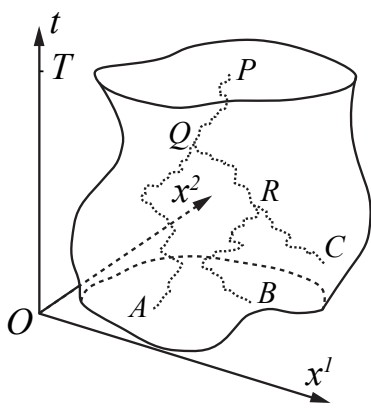

**Figure 1.** Schematic illustration of process splittings in a $t \otimes \mathbb{R}^2$ space. A stochastic process travels backward in time from point $P$ and splits into two at point $Q$, where its projected functional value is found to be sufficiently large (see text for exact meaning). One child process splits again at point $R$ where its projected functional value is found to be even larger. The independent child processes would eventually stop either in $\bar{\Omega}$ as at points $A$ and $B$, or on $\partial_1 \Omega$ as at point $C$.

into batches (Zheng, 2015). Although the probabilistic distribution of individual $\widetilde{\mathcal{F}}^{t,\boldsymbol{x}}$ is generally far from normal and largely unknown, that for the batch-wise mean of $\widetilde{\mathcal{F}}^{t,\boldsymbol{x}}$ approaches a Gaussian for a large enough sample number per batch due to the central limit theorem, and thereby a confidence interval can be calculated for the batch-wise means using the Student t-distribution (e.g., Kloeden and Platen, 1992). We use this confidence interval as an approximation to $\epsilon$. In this way, UBER

adaptively stops simulating more batches of stochastic processes when the estimated error meets a prescribed tolerance.

     In typical radiation belt problems, the functional values from various stochastic paths may differ by orders of magnitude, hence their contributions to the arithmetic mean also differ by orders of magnitude, whereas their computational efforts are of the same order. Therefore, straightforward calculation of their arithmetic mean could result in extremely slow convergence with $N$ and squander computational power. To reduce statistical variance in this procedure, a process-splitting technique

is developed based on the idea of importance sampling, i.e., to make "denser" sampling in more important "regions". In conventional Monte Carlo methods, the "region" is an "area" in a parameter space, and importance sampling effectively splits one sample point therein that would have made a huge contribution to the calculation into many sample points nearby, while weights of these samples are reduced accordingly to keep the probabilistic distribution of samples unbiased (e.g., Press et al., 1992). But unlike conventional Monte Carlo methods, the samples in the SDE method are paths which belong to a functional

space. To still implement this idea, we split the stochastic path when it is projected to contribute a large functional value.

     Fig. 1 gives an illustration of this technique in a $t \otimes \mathbb{R}^2$ space. As a stochastic path being integrated from point $P$, the functional value of the entire path (from $s = 0$ to $s = t$) is continuously predicted based on the partial path that has been realized. This projected functional value is compared to the value of some quantile (e.g., the 80th percentile) statistically





derived from all previously completed stochastic paths starting from the same position. When at some place $Q$, the projected

functional value falls above this quantile, the stochastic process is deemed to make a significant contribution to the arithmetic

mean. It is then split into a number of child processes at $Q$, and each child process traces down an independent path thereafter.

These child paths, together with their common parent path segment $PQ$, hence constitute "nearby samples" in the functional

space. This procedure can be further iterated if the projected functional value later falls into an even higher quantile (e.g., the

90th percentile), as shown at $R$. After all procedures finished, the eventual result is a tree structure of stochastic paths rooted

at $P$. For the illustration in Fig. 1, the actual functional value of the path $PQA$ will be weighted by $1/2$, and those of $PQRB$

and $PQRC$ will be weighted by $1/4$, when calculating their contributions to the mean. In the UBER code, a practical choice

for the number of children at each splitting is $4$, and that for the upper limit of offspring generations is $3$, so that a stochastic

process can be split into a maximum of $4^3 = 64$ processes. Effects of the process-splitting technique are studied in the first

problem in the next section.

It still remains to find a method to project the functional value of a stochastic path when it is only partially realized. For

this purpose, we insert a break point at $s = s' \in (0,t)$ to the integrations in Eq. (9) and see how it transforms. We simplify

the situation by only considering the stochastic processes stopping in $\bar{\Omega}$ for the moment, and denote the following functional

integrals:

$$\mathcal{U}_0^t = \exp\left(\int_0^t c\,ds - \int_0^t \lambda\,dk_s\right), \tag{17}$$

$$\mathcal{V}_0^t = \int_0^t u\exp\left(\int_0^s c\,dr - \int_0^s \lambda\,dk_r\right)ds, \tag{18}$$

in which the integrand functions $c$, $\lambda$ and $u$ are as those in Eq. (9). Then, the functional $\mathcal{F}^{t,\boldsymbol{x}}$ with the above presumptions and

notations is transformed as

$$
\begin{aligned}
\mathcal{F}^{t,\boldsymbol{x}} =\ & g_0(\boldsymbol{X}_t)\mathcal{U}_0^t + \mathcal{V}_0^t \\
=\ & g_0(\boldsymbol{X}_t)\mathcal{U}_0^{s'}\mathcal{U}_{s'}^t + \left(\mathcal{V}_0^{s'} + \mathcal{U}_0^{s'}\mathcal{V}_{s'}^t\right) \\
=\ & \left[g_0(\boldsymbol{X}_t)\mathcal{U}_{s'}^t + \mathcal{V}_{s'}^t\right]\mathcal{U}_0^{s'} + \mathcal{V}_0^{s'} \\
=\ & \mathcal{F}^{t-s',\boldsymbol{X}_{s'}}\mathcal{U}_0^{s'} + \mathcal{V}_0^{s'}, \tag{19}
\end{aligned}
$$

where $\mathcal{F}^{t-s',\boldsymbol{X}_{s'}}$ is the functional for a stochastic process that starts from the break point $(t-s',\boldsymbol{X}_{s'})$ and continues till $s = t$.

Suppose that a partial path has been realized up to $s = s'$, from it we can readily evaluate $\mathcal{U}_0^{s'}$ and $\mathcal{V}_0^{s'}$ in Eq. (19), and

therefore need an estimated $\bar{\mathcal{F}}^{t-s',\boldsymbol{X}_{s'}}$ to project the functional value $\bar{\mathcal{F}}^{t,\boldsymbol{x}}$, where a bar is put over all unrealized entities.

Specifically, we would need these three estimates: $\bar{\boldsymbol{X}}_t$, $\bar{\mathcal{U}}_{s'}^t$ and $\bar{\mathcal{V}}_{s'}^t$. In principle, a good estimation of $\bar{\boldsymbol{X}}_t$ is given by integrating

along the streamline of the $\boldsymbol{b}(t-s,\boldsymbol{x})$ field through $\boldsymbol{X}_{s'}$ till $s = t$, that is, projecting for $\bar{\boldsymbol{X}}_t$ along the ballistic trajectory of

motion while ignoring all the stochasticity since the Wiener process has zero mean. However, this integration is not much

cheaper than the realization of $\mathcal{F}^{t-s',\boldsymbol{X}_{s'}}$ itself, and thus is unaffordable. In anticipation that the total time length $t$ would not

be too large, especially when using a solution grid, and that $\boldsymbol{b}(t-s,\boldsymbol{x})$ would not vary drastically in this time interval, mapping





$\bar{\boldsymbol{X}}_t$ along the constant vector $\boldsymbol{b}(t - s', \boldsymbol{X}_{s'})$ is a good enough but much cheaper approximation. If $\bar{\boldsymbol{X}}_t$ is mapped out of $\bar{\Omega}$ so
that $g_0(\bar{\boldsymbol{X}}_t)$ is unable to be evaluated, the particular stochastic process is then disabled from splitting.

The functional values $\bar{\mathcal{U}}_{s'}^t$ and $\bar{\mathcal{V}}_{s'}^t$ are estimated by assuming that, for all possible stochastic paths belonging to the same
solution point, there exist mean functions $\bar{c}$, $\bar{\lambda}$ and $\bar{u}$ that are independent of time, and that the mean local time is proportional
to the total time length of the stochastic process, so that $\bar{k}_s = \bar{k}s$ with $\bar{k}$ the proportionality constant. Under these assumptions,
$\bar{\mathcal{U}}_{s'}^t$ and $\bar{\mathcal{V}}_{s'}^t$ can be expressed by

$$
\begin{aligned}
\bar{\mathcal{U}}_{s'}^t &= \exp\left( \bar{c} \int_{s'}^t ds - \bar{\lambda} \int_{s'}^t dk_s \right) \\
&= \exp\left[ (\bar{c}t - \bar{\lambda}\bar{k}t) - (\bar{c}s' - \bar{\lambda}\bar{k}s') \right] \\
&= \frac{\bar{\mathcal{U}}_0^t}{\exp\left( \dfrac{s'}{t} \ln\bar{\mathcal{U}}_0^t \right)},
\end{aligned}
\tag{20}
$$

and

$$
\bar{\mathcal{V}}_{s'}^t = \frac{\bar{\mathcal{V}}_0^t}{\bar{\mathcal{U}}_0^t - 1}\left[ \bar{\mathcal{U}}_0^t - \exp\left( \frac{s'}{t} \ln\bar{\mathcal{U}}_0^t \right) \right],
\tag{21}
$$

if $\bar{\mathcal{U}}_0^t \neq 1$, or by

$$
\bar{\mathcal{U}}_{s'}^t = 1,
\tag{22}
$$
$$
\bar{\mathcal{V}}_{s'}^t = \bar{\mathcal{V}}_0^t\left( 1 - \frac{s'}{t} \right),
\tag{23}
$$

if $\bar{\mathcal{U}}_0^t = 1$. The values of $\bar{\mathcal{U}}_0^t$ and $\bar{\mathcal{V}}_0^t$ can be well estimated respectively by the medians of $\widetilde{\mathcal{U}}_0^t$ and $\widetilde{\mathcal{V}}_0^t$ that are obtained from
all previously completed stochastic paths. Medians are preferred to means here because the probabilistic distributions of these
functionals are usually very skewed and heavy-tailed. This projection mechanism would become statistically more accurate as
more stochastic processes having been simulated.

## 4    Example Problems

Three example problems are provided in this section. In the first problem, we solve a Fokker-Planck equation with two source
terms, one proportional to the unknown function and the other independent of the unknown function, in both spherical coor-
dinate system and Cartesian coordinate system. Effects of the process-splitting technique are analyzed in this example. In the
second problem, an advection-dominated Fokker-Planck equation is considered. We further show that, even for a pure advec-
tion equation, the UBER code still gives the correct solutions, although it is not designed for such an equation and may not be
the most efficient method. In the last problem, we simulate the Earth's inner radiation belt by solving its Boltzmann equation
involving realistic pitch-angle diffusion and CRAND source. The treatment of periodic boundary conditions is illustrated in
this example.





### 4.1 Problem 1: Neutron Generation and Diffusion in Nuclear Material

In this problem, we consider the diffusion and generation of neutrons in a spherical nuclear material at detonation, with an initially injected Gaussian neutron distribution from a small source at the center, and a neutron-reflecting coat that allows only one half of the surface neutrons to escape. In a spherical coordinate system, the equation, initial condition and boundary conditions are (Serber, 1992)

$$\frac{\partial f}{\partial t} = \frac{1}{r^2}\frac{\partial}{\partial r}\left(r^2 D \frac{\partial f}{\partial r}\right) + Sf + v(r), \tag{24}$$

$$f(0,r) = \exp\left(-\frac{r^2}{0.02}\right), \tag{25}$$

$$\left.\frac{\partial f}{\partial r}\right|_{r=0} = 0, \tag{26}$$

$$\left.\left(D\frac{\partial f}{\partial r} + \frac{1}{2}f\right)\right|_{r=1} = 0, \tag{27}$$

where $f$ is neutron density, the constant diffusion coefficient $D = 0.1$, the constant rate of neutron generation from chain reaction $S = 2.5$, and $v(r) = 10^{-6}/(1+r)$ characterizes a weak source of neutrons spontaneously emitted in the material. The values and functional forms of these coefficients are solely designed for demonstration purpose and are not meant to be experimentally accurate.

UBER solutions are obtained at four time stamps, and are compared with those from a staggered-grid finite difference method (e.g., Ames, 2014), as shown in Fig. 2a. A turning point is observed in the solutions at $T = 0.05$, which marks the transition of the dominating neutron source from chain reaction at high background density to spontaneous emission at low density. As time goes by, effect of the spontaneous emission is overwhelmed by the fast-growing chain reaction. Even though the solutions span 8 orders of magnitude, the UBER results are virtually identical to the finite difference ones, and statistical fluctuation which is a typical feature in Monte Carlo methods is not observed in these solutions due to the adaptive algorithms and the variance reduction technique.

To demonstrate UBER's ability in multiple dimensions with a complicated boundary geometry, the same problem is also solved in a three-dimensional Cartesian coordinate system along a sphere radius. In this coordinate system, the diffusion coefficient becomes a rank-3 tensor with each diagonal component equal to $D$, and the boundary condition in Eq. (27) is applied to the only boundary that is a sphere with unit radius. The solutions are over-plotted in Fig. 2a. Consistence between the one-dimensional and the three-dimensional results is quite evident.

To analyze the effects of the process-splitting technique, we repeated the three-dimensional solutions at $T = 0.05$, but with a fixed number of stochastic processes (2048 samples per batch, 200 batches) for each solution point and with various upper limits of the offspring generations $\nu$. $\nu = 0$ indicates that process-splitting technique is disabled. For a fixed number of samples, the relative error of a solution is proportional to the square-root of the variance of sampled functional values, and determines how fast the calculation of expectation converges. The relative errors as functions of $r$ are plotted against the left y-axis of Fig. 2b, and each curve is in fact formed by the medians from eight independent and identical numerical experiments to be more statistically representative. In the range $0.4 < r < 0.9$, the relative errors are consistently reduced with higher

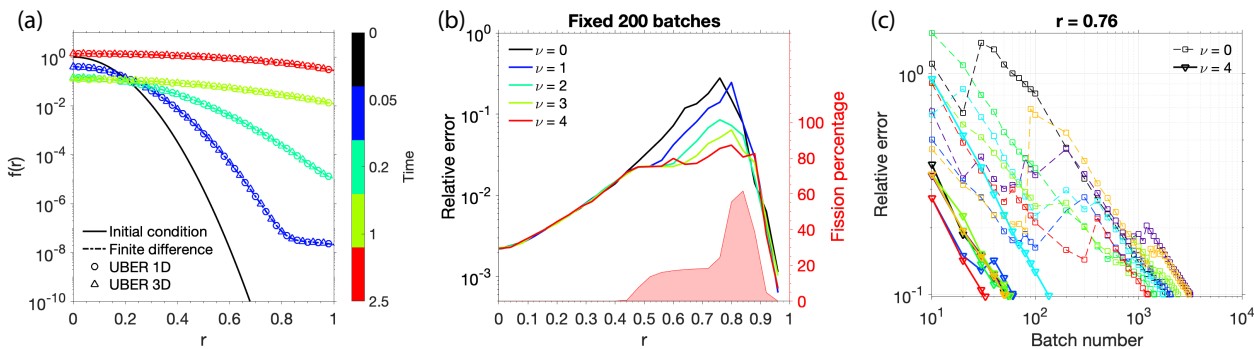

**Figure 2.** (a) UBER and finite difference solutions (dashed line) to the problem in Eqs. (24)-(27). The UBER 1D solutions (circles) are obtained in a one-dimensional spherical coordinate system, and the UBER 3D solutions (triangles) are obtained in three-dimensional Cartesian coordinates along a sphere radius. (b) Left y-axis: The relative errors of the UBER 3D solutions at $T = 0.05$, respectively obtained with the same total number of stochastic processes (2048 per batch) but different upper limits of offspring generations ($\nu$) in the process-splitting technique. $\nu = 0$ means the process-splitting is turned off. Right y-axis: The percentage of stochastic processes undergone splitting for $\nu = 4$. (c) The reduction of relative errors with increasing number of stochastic processes at the slowest converging solution point ($r = 0.76$), for $\nu = 0$ (dashed line and squares) and 4 (solid line and triangles). Colors denote different numerical experiments.

offspring generations. At the slowest converging point $r = 0.76$, the process-splitting technique with a maximum of 4 offspring generations could reduce the relative error by an order of magnitude compared to that without splitting. For this curve ($\nu = 4$), the percentages of stochastic processes undergone splitting are plotted as shaded area against the right y-axis. For $r < 0.4$, the relative errors are small and computational convergence is fast enough, process-splitting is automatically suppressed by the code to achieve an optimal speed. When the relative errors would have been large, usually a small fraction of split stochastic processes could be rather effective.

To further reveal the behavior of the process-splitting technique, Fig. 2c plots how the relative error reduces with increasing number of samples ($N$) in the Monte Carlo procedure for the solution point at $r = 0.76$. There are eight independent and identical numerical experiments respectively for $\nu = 0$ and 4, and each line represents the results from one numerical experiment. The general trend is that the relative error reduces linearly in a log-log scale plot, resembling its dependence on $N^{-1/2}$. However, without process-splitting, the relative error often jumps up sharply due to the occurrence of a very low probability sample that made a very large contribution, which severely slows down the computational convergence. With process-splitting, such



**Table 2.** Normalized wall clock time versus maximum offspring generations ($\nu$) for the UBER 3D solutions at $T = 0.05$

| $\nu$ | Normalized wall clock time* |
|---|---|
| 0 | 1 |
| 1 | 0.40 |
| 2 | 0.18 |
| 3 | 0.13 |
| 4 | 0.13 |

\* Median value from eight independent numerical tests.

jumps are largely avoided; and on average, the code uses just a little more than $1/100$ of the samples without process-splitting to achieve the same relative error of $0.1$.

In practical UBER usage, solutions are achieved with a prescribed tolerance of relative error and an adaptive number of samples. Therefore, the fast convergence with process-splitting technique could save a significant amount of computational effort even with its extra computational burden. Table 2 lists the normalized wall clock time consumed by UBER for obtaining the solution curve in three dimensions at $T = 0.05$ with a relative error tolerance of $0.1$ and a range of maximum offspring generations in process-splitting. Again, each of these numbers is the median from eight independent and identical numerical experiments. With $\nu = 3$ and $4$, the code is nearly an order of magnitude faster than that without process-splitting. The same wall clock time in these two cases indicates that the faster convergence with more offspring generations starts to be traded off by the computational overhead associated with more complicated splitting, and therefore further increasing $\nu$ would not be optimal.

### 4.2 Problem 2: Magnetized Plasma Evolution Under Instability

In the second problem, we consider a Fokker-Planck equation for the pitch-angle distribution of a magnetized plasma (e.g., Dendy, 1990). Suppose that the electrons are initially in a $\sin^2(x)$ background pitch-angle distribution where $x$ is the pitch angle. An electron beam is injected into the system centered at pitch angle $x = 0.4$. In addition to pitch-angle diffusion, the injected beam excites some kind of plasma instability that kinetically transports the distribution toward $\pi/2$ pitch angle. The equation, initial condition and boundary conditions are written as:

$$\frac{\partial f}{\partial t} = \frac{1}{G}\frac{\partial}{\partial x}\left[GD(t,x)\frac{\partial f}{\partial x}\right] - \frac{1}{G}\frac{\partial}{\partial x}\left[Gh(x)f\right], \tag{28}$$

$$f(0,x) = \sin^2(x) + \exp\left[-\frac{(x-0.4)^2}{0.02}\right], \tag{29}$$

$$f|_{x=0.05} = 0, \tag{30}$$

$$\left.\frac{\partial f}{\partial x}\right|_{x=\pi/2} = 0, \tag{31}$$

where $f$ is the electron distribution function, the Jacobian determinant $G = \sin(x)$, the diffusion coefficient $D(t,x) = (1/2\pi^2)\,\mathrm{erf}(t/2)[1+\sin^2(2x)]$, and the advection coefficient $h(x) = \cos(x)$. Note that, in most of the $x$ range, the advection coefficient is about an

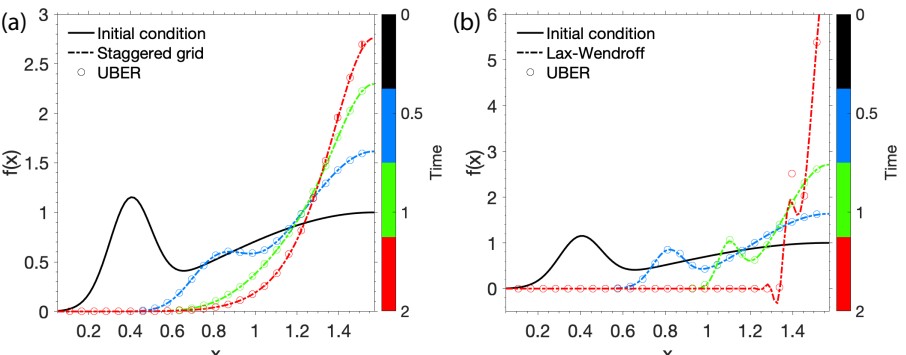

**Figure 3.** (a) UBER (circles) and staggered-grid finite difference (dashed line) solutions to the problem in Eqs. (28)-(31). (b) UBER (circles) and Lax-Wendroff (dashed line) solutions to the same problem but with zero diffusion.

order of magnitude larger in value than the diffusion coefficient. Eq. (30) indicates a loss cone at pitch angle $x = 0.05$. UBER

solutions for this problem are plotted in Fig. 3a as circles, and are in excellent agreement with those from the staggered-grid finite difference method. In these solutions, the beam evolves toward $x = \pi/2$ because of the kinetic advection. As the system relaxes, the beam eventually merges into the background, and a final stable distribution is then approached.

   Eq. (28) degenerates to a continuity equation if pitch-angle diffusion is turned off by setting $D(t,x)$ to zero. Even for such a pure advection problem, UBER can still obtain accurate and robust solutions as compared to the widely used Lax-Wendroff

method (e.g., Ames, 2014), as shown in Fig. 3b. Before $T = 2$, an advection of the beam toward $x = \pi/2$ is seen in the solutions without dispersion, and UBER results are almost identical to the Lax-Wendroff ones. The system, however, is unstable due to the positive advection velocity at $x < \pi/2$ and the zero advection velocity at $x = \pi/2$, so that the electron distribution will be piled up near $x = \pi/2$ and ultimately evolve into a singularity. For this reason, the Lax-Wendroff method begins to fail at $T = 2$ by generating unphysical negative solutions near $x = 1.3$, and will be divergent henceforth. UBER nonetheless gives the

correct results that still resolve the peak height and position of the beam.

### 4.3  Problem 3: Earth's Inner Radiation Belt Simulation

In the last problem, we demonstrate UBER's ability to solve a radiation belt Boltzmann equation by performing an inner radiation belt simulation involving both the stably trapped (out of the drift loss cone) and the quasi-trapped (in the drift loss cone) electron populations. Inspired by Xiang et al. (2020), we consider the 304-keV electrons at McIlwain's $L_M = 1.25$,

which are subject to pitch-angle scattering caused by Coulomb collisions with upper atmospheric neutrals and ionospheric ions





and electrons. The equation, initial condition and boundary conditions are:

$$\frac{\partial \bar{f}}{\partial t} + \omega_d \frac{\partial \bar{f}}{\partial \varphi} = \frac{1}{G} \frac{\partial}{\partial \alpha_0} \left( G D_{\alpha\alpha} \frac{\partial \bar{f}}{\partial \alpha_0} \right) + \frac{S_e}{p^2}, \tag{32}$$

$$\bar{f}(0, \varphi, \alpha_0) = 0, \tag{33}$$

$$\bar{f}|_{\varphi=0} = \bar{f}|_{\varphi=2\pi}, \tag{34}$$

$$\bar{f}|_{\alpha_0=\alpha_L} = 0, \tag{35}$$

$$\left. \frac{\partial \bar{f}}{\partial \alpha_0} \right|_{\alpha_0=\pi/2} = 0. \tag{36}$$

In Eq. (32),

$$\omega_d = \frac{3cLR_E}{e\mu_E} \frac{p^2}{m_e} \frac{D(\sin\alpha_0)}{T(\sin\alpha_0)} \tag{37}$$

is the drift frequency evaluated using dipole-field approximation (Schulz, 1991), in which $c$ is the speed of light in vacuum, $L$ is dipole L-shell, $R_E$ is the radius of Earth, $e$ is the elementary charge, $\mu_E$ is the magnetic moment of Earth's intrinsic dipole field, $m_e$ is electron mass, $p$ is electron momentum, $\alpha_0$ is electron equatorial pitch angle, and the functions $D(\sin\alpha_0)$ and $T(\sin\alpha_0)$ are bounce motion integrals in dipole field that are given in Schulz (1991, pp. 205-210). For simplicity, we ignore the dependence of $\omega_d$ on drift phase $\varphi$, so that the drift phase becomes equivalent to geomagnetic longitude. The Jacobian determinant $G = T(\sin\alpha_0)\sin(2\alpha_0)$. The bounce-averaged pitch-angle diffusion rate is empirically given by

$$D_{\alpha\alpha} = 10^{-5} \exp\left\{ 92.55 \left[ \cos^4\alpha_0 - \cos^4\alpha_L(\varphi) \right] \right\} + 10^{-9} \quad (\text{s}^{-1}), \tag{38}$$

which features quantitative resemblance with that calculated by realistic atmosphere and ionosphere models in Xiang et al. (2020). In this expression, $\alpha_L(\varphi)$ is the bounce loss cone angle dependent on geomagnetic longitude that is determined by drift-shell tracing in the International Geomagnetic Reference Field (IGRF, Finlay et al., 2010). $D_{\alpha\alpha}$ as a function of $\varphi$ and $\alpha_0$ is plotted in Fig. 4a: it is only significant near the bounce loss cone and in the South Atlantic Anomaly (SAA) centered at about $20°$ geomagnetic longitude, due to the closer proximity of the drift shell to the upper atmosphere in these regions. The CRAND source rate $S_e/p^2$ is approximated by (Lenchek et al., 1961; Selesnick, 2015)

$$\frac{S_e}{p^2} \approx 1.7 \times 10^{-12} \frac{(E_{max} - E)^2}{L^{2.7} \sin\alpha_0} \quad (\text{c}^3\text{cm}^{-3}\text{MeV}^{-3}\text{s}^{-1}), \tag{39}$$

where $E_{max}$ is the maximum kinetic energy (782 keV) available to electrons from neutron $\beta$-decay and $E$ is the electron kinetic energy in question, both are measured in unit of the electron rest energy (511 keV), and $L$ is the dipole L-shell, which is a variable dependent on geomagnetic longitude due to multipoles of the Earth's magnetic field. Fig. 4b plots the CRAND source rate as well as the dipole L-shell values corresponding to McIlwain's $L_M = 1.25$ obtained from drift-shell tracing in IGRF, which vary from less than $1.2$ in the SAA to above $1.3$ near $180°$ geomagnetic longitude.

Eq. (34) specifies the periodic boundary condition for the drift phase $\varphi$. In the UBER code, the periodic boundary condition is not really considered a boundary condition; rather, it is dealt with by extending the computational domain to include multiple periods, so that the Itô stochastic processes would not move out of the domain within the given time duration, except for



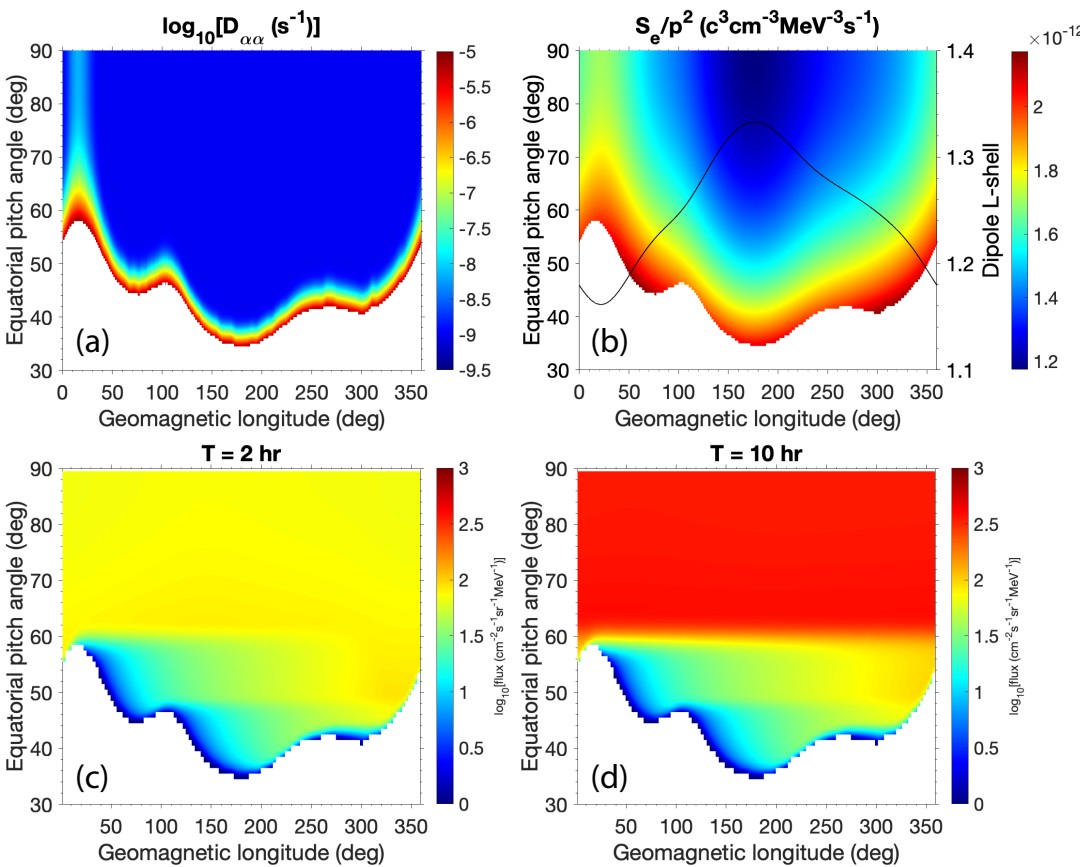

**Figure 4.** (a) Bounce-averaged pitch-angle diffusion coefficient $D_{\alpha\alpha}$ (s$^{-1}$) for 304-keV electrons. Blank area is in the bounce loss cone. (b) CRAND electron source rate $S_e/p^2$ (c$^3$cm$^{-3}$MeV$^{-3}$s$^{-1}$) for 304-keV electrons. Black line plots the variation of dipole L-shell versus geomagnetic longitude against the right y-axis, corresponding to the McIlwain's $L_M = 1.25$. (c) Calculated electron fluxes (cm$^{-2}$s$^{-1}$sr$^{-1}$MeV$^{-1}$) at $T = 2$ hours. (d) Calculated electron fluxes (cm$^{-2}$s$^{-1}$sr$^{-1}$MeV$^{-1}$) at $T = 10$ hours.

stopping on other first type boundaries. For this specific problem, the time stamp for obtaining solutions is every 2 hours, and the 304-keV electrons drift eastwards with drift periods a little longer than 2 hours. Therefore, the computational domain is

extended for one extra period of $\varphi$ from 0 to $-2\pi$ since the stochastic processes retrograde in time. However, solutions are only sought in the right half of the domain for $\varphi$ between 0 and $2\pi$ at each time stamp, and after that, they are copied to the left half to form the entire initial condition for the next time stamp.

The simulation is performed with an initially empty radiation belt as indicated by Eq. (33), and electrons are gradually generated by the CRAND source and meanwhile lost to the bounce loss cone. Fig. 4c and 4d show the solution electron

fluxes calculated from $\bar{j} = \bar{f}p^2$ after 2 hours and 10 hours, respectively. The characteristic west-east electron flux gradient is formed for the quasi-trapped population ($\alpha_0 < 60°$) within the first 2 hours, and changes very little over time because the SAA sweeps these electrons out every drift period. Weak pitch-angle diffusion of electron fluxes from the quasi-trapped population





toward the stably trapped population can be observed at $T = 2$ hours when the stably trapped fluxes are still low, due to the stronger source rate in the quasi-trapped region. At $T = 10$ hours, direction of the pitch-angle diffusion is reversed. Even with

355 atmospheric loss, the CRAND source is strong enough to continuously contribute to the trapped electron fluxes, which are increased by one order of magnitude in 8 hours.

## 5 Conclusion and Discussion

In conclusion, we have built a numerical solver for the general form of kinetic equations that appear in radiation belt studies. Based on the SDE method, the solver is coded to work in arbitrarily provided coordinate systems up to three dimensions, with

360 user-specified boundary geometry, boundary conditions, and equation terms. We have also designed adaptive algorithms and a variance reduction technique for the SDE method, which had enhanced its computational speed by one order of magnitude in our test. The example problems in this article demonstrated the solver's versatility and robustness in dealing with a range of problems that might each require a different solver in other methods. The solver, named UBER, has been programmed into a FORTRAN library that can be easily incorporated with other more complicated space physics models.

Several other forms of radiation belt kinetic equation should also be solvable by the method presented in this article. In formulating the Boltzmann equation (2), we have assumed that the unperturbed particle Hamiltonian $H_0$ is independent of phases of particle motion. For lower-energy ring current particles, the convective electric field potential energy is not negligible in their Hamiltonian, and therefore $H_0$ would be dependent on the drift phase. As such, expanding the Poisson bracket $[\bar{f}, H_0]$ on the left-hand side of the Boltzmann equation will result in additional terms involving partial differentials with respect to

the generalized momenta $Q^\iota$ (cf. Appendix A). For a radiation belt model including ring current particles, the general form of Boltzmann equation will be

$$\frac{\partial \bar{f}}{\partial t} + \dot{\varphi}^\iota \frac{\partial \bar{f}}{\partial \varphi^\iota} + \dot{Q}^\iota \frac{\partial \bar{f}}{\partial Q^\iota} = \dots, \qquad (40)$$

in which the omitted right-hand side is exactly the same as that of Eq. (2). The Boltzmann equations of the so-called four-dimensional radiation belt models, such as the CIMI model (Fok et al., 2014), the VERB-4D model (Aseev et al., 2016), and the K2 MHD-particle model (Elkington et al., 2019) are of this type. Similar to the treatment of Eq. (2), Eq. (40) can be

obtained from Eq. (1) by introducing the new coordinates $x^\xi = \{Q^\mu, Q^\iota, \varphi^\iota\} \equiv \{x^\mu, x^\kappa\}$, which enlarges the index set from $\alpha$ to $\xi$, and performing the following transformation of equation coefficients:

$$\begin{cases} \widetilde{D}^{\mu\nu} = D^{\mu\nu}, \quad \widetilde{D}^{\xi\kappa} = 0, \\ \widetilde{h}^\xi = \{h^\mu, \dot{x}^\kappa\}, \\ \widetilde{S} = S + \partial_\kappa \dot{x}^\kappa + \dot{Q}^\iota \partial_\iota \ln G, \\ \widetilde{v} = v. \end{cases} \qquad (41)$$

Therefore, the Boltzmann equation (40) can also be solved by the method presented in this article in principle. However, such four-dimensional simulations are beyond the current scope of the UBER code since it is only coded for up to three dimensions in space.





Nonlinear evolution of phase space density occurs when the particle scatterings are not only small-scale but also large-scale, usually as a result of trapping by intense plasma waves (e.g., Bortnik et al., 2008; Albert et al., 2013). In this case, the right-hand side of the kinetic equation must include terms of non-local transport of phase space density by these large-scale scatterings, and the equation is formulated as (Artemyev et al., 2016; Zheng et al., 2019)

$$\frac{\partial \bar{f}}{\partial t} = \frac{1}{G}\frac{\partial}{\partial Q^\alpha}\left(G D^{\alpha\beta}\frac{\partial \bar{f}}{\partial Q^\beta}\right) - \frac{1}{G}\frac{\partial}{\partial Q^\alpha}\left(G h^\alpha \bar{f}\right) - \left(\int P_{Q\to\widetilde{Q}}\widetilde{G}d\widetilde{Q}^\alpha\right)\bar{f} + \int P_{\widetilde{Q}\to Q}\widetilde{f}\widetilde{G}d\widetilde{Q}^\alpha, \tag{42}$$

in which $\widetilde{f}$ is a shorthand for the function $\bar{f}(t,\widetilde{Q}^\alpha)$, and $\widetilde{G}$ is the Jacobian determinant evaluated at $\widetilde{Q}^\alpha$. With nonlinear wave-particle interactions, phase bunching effect gives rise to the advection characterized by the coefficients $h^\alpha$. The function $P_{Q\to\widetilde{Q}}$ is the trapping probability density per unit time from $Q^\alpha$ to $\widetilde{Q}^\alpha$, that is, particles are trapped by the wave field at $Q^\alpha$ and subsequently escape from trapping at $\widetilde{Q}^\alpha$, and is considered a known function which can be evaluated from single particle behaviors by either perturbation theory of Hamiltonian mechanics (e.g., Artemyev et al., 2016) or test-particle simulations (e.g., Vainchtein et al., 2018). Note that, since the unknown function is contained in the last integral term, Eq. (42) is an integro-differential equation. However, formal similarity between Eq. (42) and the Fokker-Planck equation (1) suggests that an asymptotic solution of Eq. (42) may be achieved by Taylor expanding $\widetilde{f}$ as

$$\widetilde{f} = \widetilde{f}_0 + \widetilde{f}_0' t + \ldots, \tag{43}$$

where $\widetilde{f}_0 = \bar{f}(0,\widetilde{Q}^\alpha)$ and $\widetilde{f}_0'$ indicates its time-derivative function evaluated at $t = 0$. When applying the SDE method with a solution grid, the functions $\widetilde{f}_0$ and $\widetilde{f}_0'$ can be obtained from solutions of previous time stamps. Then, by defining the following coefficients

$$S(Q^\alpha) = -\int P_{Q\to\widetilde{Q}}\widetilde{G}d\widetilde{Q}^\alpha, \tag{44}$$

$$v(t,Q^\alpha) = \int P_{\widetilde{Q}\to Q}\widetilde{f}_0\widetilde{G}d\widetilde{Q}^\alpha + t\int P_{\widetilde{Q}\to Q}\widetilde{f}_0'\widetilde{G}d\widetilde{Q}^\alpha + \ldots, \tag{45}$$

which are now known functions, Eq. (42) is transformed into the form of Eq. (1), and is readily solvable by the UBER code. In this way, the simulations of nonlinear wave-particle interactions in the radiation belts could hence be unified with the well-developed simulations in the quasi-linear theory.

*Code and data availability.* The UBER library is free and open source. The current version of UBER is available from the GitHub repository https://github.com/zheng-lh/UBER (last access 14 April 2021) under the MIT license. The exact version of the UBER library used to produce the results used in this paper is archived on Zenodo (https://doi.org/10.5281/zenodo.4671646, Zheng, 2021), as are input data and scripts to run the library and produce the plots and tables for all the simulations presented in this paper (https://doi.org/10.5281/zenodo.4686050, Zheng et al., 2021).





## Appendix A: A Formal Derivation of Radiation Belt Kinetic Equations

We consider, for simplicity, a hypothetical radiation belt whose particle motion has two well-separated periods which define
two pairs of action-angle variables $\{\varphi^1, \varphi^2, J^1, J^2\}$. We assume that the phase angle $\varphi^1$ changes much faster than $\varphi^2$, and
hence call $\{\varphi^1, J^1\}$ the fast variables and $\{\varphi^2, J^2\}$ the slow variables. We further assume, for a moment, that the Hamiltonian
of particle motion

$$H(\varphi^1, \varphi^2, J^1, J^2, t) = H_0(\varphi^2, J^1, J^2, t) + \delta H(\varphi^1, J^1, J^2, t) \tag{A1}$$

is constituted of an unperturbed part $H_0$ that depends on the slow phase and a perturbation $\delta H$ that is caused by electromagnetic
forces whose variation time scale is shorter than the periodicity $2\pi/\dot{\varphi}^1$. Apparently, $\delta H$ is a periodic function of $\varphi^1$. Upon
averaging over $\varphi^1$, the perturbation cancels out, so that

$$\langle \delta H \rangle \equiv \frac{1}{2\pi} \int_0^{2\pi} \delta H d\varphi^1 = 0. \tag{A2}$$

These presumptions allow the particle phase space density

$$f(\varphi^1, \varphi^2, J^1, J^2, t) = \bar{f}(\varphi^2, J^1, J^2, t) + \delta f(\varphi^1, J^1, J^2, t) \tag{A3}$$

to be so decomposed into a fast-phase-averaged part $\bar{f} \equiv \langle f \rangle$ and a perturbation $\delta f$, also periodic in $\varphi^1$, which has $\langle \delta f \rangle = 0$
by definition.

For a collisionless plasma, evolution of $f$ is governed by the Vlasov equation

$$\frac{\partial f}{\partial t} + \dot{\varphi}^\alpha \frac{\partial f}{\partial \varphi^\alpha} + \dot{J}^\alpha \frac{\partial f}{\partial J^\alpha} = 0, \tag{A4}$$

in which the index $\alpha = 1, 2$. Expressing $\dot{\varphi}^\alpha$ and $\dot{J}^\alpha$ by Hamilton's canonical equations, the Vlasov equation can be expanded
in light of (A1) and (A3). When averaging the expanded equation over $\varphi^1$, all terms to the first order in perturbation vanish
due to either their null phase average or periodicity in $\varphi^1$, and the remaining terms form the equation

$$\frac{\partial \bar{f}}{\partial t} + \frac{\partial H_0}{\partial J^2} \frac{\partial \bar{f}}{\partial \varphi^2} - \frac{\partial H_0}{\partial \varphi^2} \frac{\partial \bar{f}}{\partial J^2} = - \left\langle \frac{\partial \delta H}{\partial J^1} \frac{\partial \delta f}{\partial \varphi^1} - \frac{\partial \delta H}{\partial \varphi^1} \frac{\partial \delta f}{\partial J^1} \right\rangle, \tag{A5'}$$

or organized into Poisson brackets with respect to the canonical coordinates $\{\varphi^\alpha, J^\alpha\}$,

$$\frac{\partial \bar{f}}{\partial t} + [\bar{f}, H_0] = - \langle [\delta f, \delta H] \rangle. \tag{A5}$$

In fact, this equation form is more neatly derived from Liouville's theorem (e.g., Goldstein, 1980, Chap. 9) which says

$$\frac{df}{dt} = \frac{\partial f}{\partial t} + [f, H] = 0. \tag{A6}$$

Phase averaging Eq. (A6) over $\varphi^1$ and noting that $\langle [\delta f, H_0] \rangle = \langle [\bar{f}, \delta H] \rangle = 0$ directly gives Eq. (A5).

Eq. (A5) appears in a form of a Boltzmann equation for the phase-averaged phase space density: the left-hand side describes
evolution of the unperturbed system in the slow variables; whereas the right-hand side, involving only the perturbed quantities





and the fast variables, serves the role of a collision integral: indeed, it can be viewed upon as "collisions" between particles and the perturbing electromagnetic field. In this regard, we symbolically denote the right-hand side $\left\langle \left( \frac{\partial f}{\partial t} \right)_w \right\rangle$ in analogy to

that caused by real collisions, with the subscript designating wave-particle interaction.

Eq. (A5) is closed when its right-hand side can be expressed in terms of $\bar{f}$ under certain approximations. If both $\delta H$ and $\delta f$ are small compared to their unperturbed counterparts, $\delta f$ can be directly solved from the linearized Vlasov equation retaining only the first-order terms in expansion, and after mathematical transformations, gives the expression (e.g., Lerche, 1968; Kaufman, 1972)

$$\left\langle \left( \frac{\partial f}{\partial t} \right)_w \right\rangle = \frac{\partial}{\partial J^1} \left( D_w \frac{\partial \bar{f}}{\partial J^1} \right), \tag{A7}$$

where the coefficient $D_w$ is a functional of $\delta H$. The corresponding theory is called the quasi-linear theory. However, when the perturbing electromagnetic wave is sufficiently coherent, $\delta f$ may become large even if $\delta H$ remains small. In this situation, $\left\langle \left( \frac{\partial f}{\partial t} \right)_w \right\rangle$ is estimated by considering particle phase trajectories near the resonance point (e.g., Artemyev et al., 2016). The result would then contain corrections to Eq. (A7) which are due to particles trapped in phase with the wave, whose formulation in the current setup could be inferred from that of Eq. (42) in the body; and we hereby do not elaborate.

Taking account collisions in the plasma would introduce to Eq. (A4) a collision term not describable by the single-particle Hamiltonian, so that the transport equation becomes the Boltzmann equation

$$\frac{\partial f}{\partial t} + \dot{\varphi}^\alpha \frac{\partial f}{\partial \varphi^\alpha} + \dot{j}^\alpha \frac{\partial f}{\partial J^\alpha} = \left( \frac{\partial f}{\partial t} \right)_c. \tag{A8}$$

Following the same treatment from (A4) to (A5), Eq. (A8) leads to the fast-phase-averaged equation

$$\frac{\partial \bar{f}}{\partial t} + [\bar{f}, H_0] = \left\langle \left( \frac{\partial f}{\partial t} \right)_w \right\rangle + \left\langle \left( \frac{\partial f}{\partial t} \right)_c \right\rangle. \tag{A9}$$

For Coulomb collisions, small-angle scatterings at large impact parameters dominate due to the long range of Coulomb force, and consequently the phase-averaged collision integral can be expanded into a Fokker-Planck form in the generalized momenta

that are changed by the collisions (Lifshitz and Pitaevskii, 1981, Chap. 2 and 4), i.e.,

$$\left\langle \left( \frac{\partial f}{\partial t} \right)_c \right\rangle = \frac{\partial}{\partial J^1} \left( D_c \frac{\partial \bar{f}}{\partial J^1} \right) - \frac{\partial}{\partial J^1} \left( h_c \bar{f} \right), \tag{A10}$$

which usually only involves the fast momentum $J^1$ on time scales much shorter than $2\pi/\dot{\varphi}^2$. The transport coefficients $D_c$ and $h_c$ are determined from the particle species and their collision cross-sections. We note again that, in the phase-averaged kinetic equation (A9), slow and fast variables are separated onto each side of the equation.

Neglecting the source and loss terms, the quasi-linear kinetic equations in the body of this paper could all be recovered

from Eqs. (A7), (A9) and (A10), which are already in the same form as Eq. (40). If there are no slow variables, the Poisson bracket on the left-hand side of Eq. (A9) vanishes, and the equation reduces to the Fokker-Planck equation (1). If there are slow variables but $\varphi^2$ is cyclic to the unperturbed Hamiltonian, the left-hand side of Eq. (A9) would then contain the first two terms shown in Eq. (A5′), which is in the form of Eq. (2). In this case, the dependence of $\bar{f}$ on $\varphi^2$ is introduced by means other than the Hamiltonian, such as the $\varphi^2$-dependent boundary geometry, boundary conditions or collision terms.



*Author contributions.* LZ developed the UBER library, conducted the benchmark experiments, and wrote the paper. LC and AAC contributed to the conceptualization of the UBER library. PW, ZX and XL provided ancillary code and data for the benchmark experiments.

*Competing interests.* The authors declare that they have no conflict of interest.

*Acknowledgements.* Liheng Zheng acknowledges support of this work by NASA grant 80NSSC18K1224. Anthony Chan acknowledges support of this work by NASA grant NNX15AI93G.



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
