# Peer review of "UBER v1.0: A universal kinetic equation solver for radiation belts"

_Geoscientific Model Development, 2021_

## Author Comment (AC1)

**Authors comments on gmd-2021-122**

On behalf of all authors of the manuscript, I would like to express our great appreciation to the referees for their careful evaluation of our manuscript and the very thoughtful comments on its improvement. The manuscript has gone through revisions per their suggestions, and the changes are highlighted in blue color in the tracked-change version. In this response, I will be referring to the line numbers in the tracked-change version.

All three referees have suggested adding discussion on the UBER code efficiency as compared to finite difference methods. In the revised manuscript, a new paragraph is added in the discussion section between Line 395–416 and a new figure (Fig. 6) is added for this purpose. The main conclusion from that paragraph is that the fully parallelized UBER code is roughly an order of magnitude slower than its serial finite-difference counterpart. Caveats to this conclusion are also discussed. In the following, I will respond to the referees' comments point by point.

**RC1**

**"I would clarify around line 20 that Js are the canonical momentums; otherwise alpha would go from 1 to 6."**

Thank you for this careful pick. In both *Goldstein* [1980] and *Landau and Lifshitz* [1976], J is named the action variable and  $\varphi$  the angle variable, and they are collectively called the action-angle variables. In obedience to this nomenclature, I have changed the inaccurate "action-integral variables" in Line 21 to "action variables", and given a reference to *Goldstein* [1980, Chap. 10] earlier in Line 16 when Hamiltonian mechanics is first mentioned.

"I thought that we neglect the phases of J simply because we assume isotropy in all phis . I could not precisely follow the argument of electric potential energy. Please elaborate."

I assume this comment is referring to the paragraph between Line 34–45. Appendix A gives a detailed explanation on the forms of the kinetic equations. There are two dependent functions on  $\varphi$  here, one is the phase space density  $\bar{f}$ , and the other is the particle Hamiltonian  $H_0$ . These two dependences are not equivalent. For example, if  $H_0$  is independent on  $\varphi$  ( $\varphi$  is a cyclic variable),  $\bar{f}$  could still be dependent on  $\varphi$  for the reasons listed in the last sentence of

Appendix A; and in this situation the corresponding kinetic equation take the form as given by Eq. (2). Conversely, if assuming  $\overline{f}$  isotropic in  $\varphi$  but  $H_0$  dependent on  $\varphi$ , the left-hand side of the kinetic equation would then be like

$$\frac{\partial \bar{f}}{\partial t} + \dot{Q}\frac{\partial \bar{f}}{\partial Q} = \dots, \tag{C1}$$

according to Eq. (A5') in Appendix A. The argument between Line 40–44 on electric potential energy explains why for radiation belt particles the term  $\dot{Q}\partial\bar{f}/\partial Q$  should vanish, i.e., why  $H_0$  should be independent on  $\varphi$ .

Writing explicitly, the particle unperturbed Hamiltonian takes the form

$$H_0 = \frac{p^2}{2m} + e\Phi(\varphi), \tag{C2}$$

in which the only dependence on drift phase  $\varphi$  comes from the electric potential energy  $e\Phi$ . Variation of  $\Phi$  along the particle drift path can be roughly estimated by the cross polar cap potential, since the magnetic field lines connecting the polar ionosphere and the drift path can be well approximated as equipotentials. For Earth, cross polar cap potential is usually a few tens of kilovolts in quiet times, and rarely reaches above 200 kV during disturbances [e.g., *Gao*, 2012]. Consequently, the electric potential energy variation is in the order of  $10^1$  keV, which is much less than the kinetic energy  $(p^2/2m)$  of radiation belt particles  $(10^2 \sim 10^4$ keV for electrons,  $10^1 \sim 10^2$  MeV for protons). Therefore, for particles in the radiation belt energy range, the term  $e\Phi$  can be neglected in  $H_0$ , resulting in its independence on  $\varphi$ , and the disappearance of  $\dot{Q}\partial\bar{f}/\partial Q$  in the kinetic equation. However, for ring current electrons,  $e\Phi$  is no longer negligible, and this case is discussed in the discussion section.

I struggled on whether to include these numerical ranges in the manuscript to make it more explicit. On a second thought, since these ranges specifically refer to Earth, whereas the introduction section talks about the general radiation belts of magnetized planets, I think it is better to leave them for the readers of different walks to digest; otherwise the introduction would become embarrassingly verbose.

"While the paper discusses in detail the advantages of this approach, it omits the disadvantages. I would suggest clearly saying how long the presented runs take to run on a regular PC and on a supercomputer. I would also specify how much wall clock computing would take to calculate one day charging along a given satellite orbit. Also, specify how much slower this approach than more traditional approaches for the multi-dimensional diffusion equations."

Thanks for these good suggestions. And they are addressed in the added paragraph between Line 395–416 to my best resources.

**"The presented tests all show examples that are initiated with smooth initial conditions. Please provide examples similar to Aseev et al., 2016 with strong gradients in initial conditions."**

This example is provided as the new example Problem 3 in the revised manuscript. The conclusion from that example is, for advection equations with constant advection velocity, the UBER code could achieve exact solutions regardless of the discontinuities and infinite gradients in the initial condition; for variant advection velocity, the UBER solution errors will be small. The reasons for these UBER code performances are also discussed there. Since now this is the example problem where the periodic boundary condition first appears, the explanation on its treatment is moved from Problem 4 to here.

**RC2**

"1, The main problem I have with the paper is that it failed to discuss the main concern of the the SDE method: its speed, especially compared with finite difference methods etc. Clearly, the SDE code with newer techniques is much much more efficient than the traditional one. But how does it compare with simple finite difference methods in the three cases discussed? I think the authors could discuss the comparison in different situations; e.g., when one cares about only a few snapshots, and when needs to know the whole history of evolution. Having a fair discussion about the disadvantage of the method does not make the paper less significant, but instead show future directions where improvements can be made."

Thank you for these suggestions. I have addressed comparisons of code efficiency in the newly added paragraph between Line 395–416 to my best resources.

"2, Lines 45-50: These listed numerical codes use either finite difference type methods or SDE/layer methods. None of the methods are limited by the choice of coordinates. Those authors chose a particular choice of coordinates probably because they did not intend to build a general library or because they tended to demonstrate a new method. I think it would nice for the authors to take this into consideration when discussing previous models."

Thanks for this thoughtful comment. The sentence in Line 48–49 is modified to take this point in consideration.

"3, Lines 53-55: The complicated geometry is a problem for some of the models, mainly because radiation belt people seem to have a preference for finite difference methods. There exist general powerful finite volume/elements methods that can handle complicated boundary geometry. So here '...would be challending for numerical methods' should really be '...would be challenging for finite difference methods.' "

Corrected.

"4, In Table 1, the UBER library input, can the current code handle time-dependent D? I know the method can, but not sure if it has been implemented by the code."

Yes, the code can, as also implied by the curved cylinder in Fig. 1. To make it more explicit, the phrase "in the UBER code" is added to the sentence in Line 142.

"5, Lines 158: No, the grids in the layer method need NOT to be uniform. It was simply chosen for simplicity for estimating error and demonstration purposes.

"6, Lines 155-159: Yes, in SDE methods, one can design this kind of method to obtain the global solutions of f and its history. However, the key to implementing this in SDE is actually about finding an appropriate interpolation method. 'The only operation ... would be interpolation' sounds like finding such an interpolation method is easy, while in fact it is NOT. For the described choice of irregular domain or nonuniform domain, the interpolation method still needs to be of high order to reduce systematic error from interpolation, and to preserve positivity of the solution. For example, if a simple 4th order polynomial interpolation method is used, one might introduce oscillation of f, and hence negative value s of f, from interpolation, and hence violates one of the key advantage of the SDE method. That is why Tao et al., 2016 (doi:10.1002/2015JA022064) introduced those higher-order positivity-preserving methods to be used with layer method."

The sentence between Line 158–160 has been modified. "The only operation" refers to that there is no grid-scale differentiation or averaging as in finite difference methods and layer methods, therefore it allows for the use of irregular grids. It also hints at the possibility to entirely leave the interpolation part to sophisticated libraries on irregular grids, such as those based on Delaunay triangulation, so that the SDE method could achieve to some extent the features of finite element methods, although the current version has not reached that level of sophistication. A convenient way to preserve positivity of f is to interpolate in its logarithm rather than f itself (except where f = 0), and then take exponential of the interpolated value, and this is how the UBER code deals with this problem now. If you'd bother diving into UBER's source code, you would find that the grid and the interpolation code does constitute a big part of it.

"7: Lines 310-315: There actually exists flux-limited Lax-Wendroff methods that can avoid introducing unphysical negative solutions. And in the case of forming steep gradients, it is well-known that one should add flux-limiters to Lax-Wendroff type methods."

Thank you for pointing out. A caveat is added to the sentence in Line 315–316.

**RC3**

"1. Could the authors clarify or discuss further if the UBER solver can handle boundary conditions at varying locations or not? If we look at the radial diffusion model as a simple

example, in the traditional solvers based on finite different method, it is often driven by datadriven outer boundary conditions at a fixed L\*. However, the L\* location of the satellite data providing the outer boundary is actually varying in time, which could lead to data gaps in the outer boundary condition and uncertainties in the model results. Discussions on if and how this type of boundary conditions can be implemented in UBER will further increase the significance of the work."

Yes, the UBER code can handle time-variable boundary locations. This is now explicitly indicated in Line 142. Boundary geometry, boundary conditions as well as other input items as listed in Table 1 are specified to the UBER library by a user-editable template file named user\_input.F90, whose usage is explained in detail in the library's README file (https://github.com/zheng-lh/UBER). The user specifies the time-variable boundary locations to the code by giving its equation  $\psi(t, x^{\alpha}) = 0$  (as a FORTRAN function) in the template file, which can be either analytical or numerical.

"2. The reviewer is also curious how the coordinate conversion between adiabatic invariants and energy and pitch angle can be implemented in UBER. As discussed in Subbotin and Shprits [JGR, 2012], several 3D diffusion radiation belt models 'utilized two grids to solve the Fokker-Planck equation; one grid, which keeps the first and second adiabatic invariants constant, was used for the computation of the radial diffusion, and the other grid, orthogonal in energy and pitch angle at each fixed radial distance, was used for the computation of energy diffusion, pitch angle diffusion, and mixed energy and pitch angle diffusion.' At each time step, the results were converted and interpolated between the two grids, which could lead to uncertainties in the model results depending on how the conversion and interpolation are performed. Is this coordinate conversion still needed in UBER? If not, why?"

This coordinate conversion is not needed in UBER. The "two grids" you mentioned here come from the operator splitting technique that finite difference models usually employ to deal with three-dimensional diffusion. Although one can write the Fokker-Planck equation in the variable set  $\{\alpha_0, E, L^*\}$ , these variables do not form a coordinate system of the phase space. The variables that constitute a coordinate system are the adiabatic invariants  $\{\mu, K, L^*\}$ . The conversion between  $\{\mu, K\}$  and  $\{\alpha_0, E\}$  at fixed  $L^*$  depends on the magnetic field model (usually assumed a dipole), and intrinsically excludes the drift-shell splitting effects which would cause diffusion in the  $\mu$ - $L^*$  and K- $L^*$  directions. Nonetheless, finite different models still use  $\{\alpha_0, E, L^*\}$  as equation variables because the computational domain is much more regular in the  $\{\alpha_0, E\}$  coordinates than in the  $\{\mu, K\}$  coordinates at a fixed  $L^*$ , which makes ease for their numerical schemes. For the UBER code, problem dimensionality and domain irregularity are not problems, therefore it can directly solve the equation in the coordinates of adiabatic invariants, and hence the coordinate conversion you mentioned is totally unnecessary for the UBER code. "3. It will greatly enhance the significance of the paper if discussions are included in comparing the efficiency, stability, and accuracy between the UBER code and traditional finite difference codes. It is demonstrated the UBER is more efficient than previous SDE codes, but how does it compare with the traditional solvers based on the finite difference method? Is it still much less efficient if global distributions of radiation belt electrons in L, energy, and pitch angle are targeted? How about if we only need to solve for the distribution locally at certain L, energy, and pitch angle? The authors could perhaps use example problem 3 to compare the efficiency, stability, and the accuracy between the two different types of solvers and then expand the discussion to higher-dimension models such as 3D diffusion models."

Thank you for these good suggestions. A discussion on UBER's efficiency as compared to finite different methods is added in the paragraph between Line 395–416 to the author's best resources, as mentioned in the very beginning of this response. In general, the SDE method, even with parallel computation, is still much slower than finite difference ones (about an order of magnitude in the tests). However this conclusion has many caveats as you mentioned here, which are further discussed in that paragraph. I hope that paragraph adequately addresses your comments.

**References**

Gao, Y. (2012), Comparing the cross polar cap potentials measured by SuperDARN and AMIE during saturation intervals, *J. Geophy. Res. Space Physics*, 117, A08325, doi:10.1029/2012JA017690.